

# Comparative analyses of putative toxin gene homologs from an Old World viper, *Daboia russelii*

Neeraja M. Krishnan and Binay Panda

Ganit Labs, Bio-IT Centre, Institute of Bioinformatics and Applied Biotechnology, Bangalore, India

## ABSTRACT

Availability of snake genome sequences has opened up exciting areas of research on comparative genomics and gene diversity. One of the challenges in studying snake genomes is the acquisition of biological material from live animals, especially from the venomous ones, making the process cumbersome and time-consuming. Here, we report comparative sequence analyses of putative toxin gene homologs from Russell's viper (*Daboia russelii*) using whole-genome sequencing data obtained from shed skin. When compared with the major venom proteins in Russell's viper studied previously, we found 45–100% sequence similarity between the venom proteins and their putative homologs in the skin. Additionally, comparative analyses of 20 putative toxin gene family homologs provided evidence of unique sequence motifs in nerve growth factor (NGF), platelet derived growth factor (PDGF), Kunitz/Bovine pancreatic trypsin inhibitor (Kunitz BPTI), cysteine-rich secretory proteins, antigen 5, andpathogenesis-related1 proteins (CAP) and cysteine-rich secretory protein (CRISP). In those derived proteins, we identified V11 and T35 in the NGF domain; F23 and A29 in the PDGF domain; N69, K2 and A5 in the CAP domain; and Q17 in the CRISP domain to be responsible for differences in the largest pockets across the protein domain structures in crotalines, viperines and elapids from the *in silico* structure-based analysis. Similarly, residues F10, Y11 and E20 appear to play an important role in the protein structures across the kunitz protein domain of viperids and elapids. Our study highlights the usefulness of shed skin in obtaining good quality high-molecular weight DNA for comparative genomic studies, and provides evidence towards the unique features and evolution of putative venom gene homologs in vipers.

## INTRODUCTION

Snake venom genes and their products offer an excellent model system to study gene duplication, evolution of regulatory DNA sequences, and biochemical diversity and novelty of venom proteins. Additionally, snake venoms have tremendous potential in the development of new drugs and bioactive compounds (*Vonk et al., 2011*). Previous studies have highlighted the importance of gene duplications and/or sub-functionalization (*Hargreaves et al., 2014*; *Malhotra et al., 2015*; *Rokyta et al., 2011*) and transcriptional/post-transcriptional mechanisms (*Casewell et al., 2014*) contributing towards snake venom

Corresponding author
Binay Panda, binay@ganitlabs.in

diversity. Venom studies, so far, have extensively used data from proteomics experiments alongside individual gene sequences or sequences of particular family members to study variations on gene structure and their sequence composition. Presently, whole genome sequences of several snake species, king cobra *Ophiophagus hannah* (*Vonk et al., 2013*); Burmese python *Python bivitattus* (*Castoe et al., 2013*); rattlesnake *Crotalus atrox* (*Dowell et al., 2016*); Florida pygmy rattlesnake *Sistrurus miliarius barbouri* (*Vicoso et al., 2013*); garter snake *Thamnophis elegans* (*Vicoso et al., 2013*); five-pacer viper *Deinagkistrodon acutus* (*Yin et al., 2016*); pit viper *Protobothops mucrosquamatus* (NCBI Accession PRJDB4386); and corn snake *Pantherophis guttatus* (*Ullate-Agote, Milinkovitch & Tzika, 2014*) have either been published or their sequence been made available in the public domain. In addition, genome sequencing efforts are either underway or the sequences of venom-associated genes have been deposited in the databases for a few others (*Kerkkamp et al., 2016*). Out of the sequenced genomes, only a few have been annotated, or made public, a key requirement for comparative analysis of genes. This, along with the lack of availability of whole genome sequences and/or complete transcript sequences from venom glands for most snakes has limited studies on toxin gene orthologies and gene variation among venomous snakes.

Four snakes, Russell's viper (*Daboia russelii*), saw-scaled viper (*Echis carinatus*), spectacled cobra (*Naja naja*), and common krait (*Bungarus caeruleus*) are responsible for most snakebite-related mortality in India (*Mohapatra et al., 2011*; *Warrell et al., 2013*; *Whitaker, 2015*). Russell's viper is a member of the taxon Viperidae and subfamily Viperinae and is responsible for large numbers of snakebite incidents and deaths in India. Very little is known about the diversity of genes from any viper, including the only viperine where complete genome sequence information is available (European adder, *Vipera berus berus*, https://www.ncbi.nlm.nih.gov/bioproject/170536). Lack of gene annotation from this viper using transcripts obtained from venom glands and other snake species reduces the scope of a detailed comparative study on genes, including the toxin-associated genes. Such a study involving various groups of venomous and non-venomous snakes, in addition to other venomous vertebrates and invertebrates, will facilitate our understanding on the evolution of these genes, their diversity, and function.

One of the challenges in studying the genomes of venomous animals is related to sample acquisition. Additionally, in India, Government permission is required to catch snakes and extract blood samples from them (all snakes are protected in India under the Indian Wildlife Protection Act, 1972). This may be partially circumvented by the use of shed skin that does not require drawing blood or taking any tissue from the animals. However, working with DNA isolated from shed skin has its own challenges. Microbial contamination, lack of full-length DNA in the shed skin cells, rapid degradation of DNA in humid conditions and computational challenges in dealing with short stretches of DNA are some of the bottlenecks for working with DNA from shed skin.

In the current study, we explored the possibility of getting putative toxin gene homolog information from skin-derived low-coverage whole-genome sequencing data from Russell's viper, and performed comparative analysis versus major toxin proteins from a previously studied report (*Sharma et al., 2015*). We used the coding sequences and annotation from a previously characterized crotaline, a pit viper, *Protobothrops mucrosquamatus* for the

analysis. On the venom homologs, we focused our analyses on five key protein domains; nerve growth factor (NGF), platelet derived growth factor (PDGF), Kunitz/Bovine pancreatic trypsin inhibitor (Kunitz BPTI), cysteine-rich secretory proteins, antigen 5, and pathogenesis-related 1 proteins (CAP) and cysteine-rich secretory protein (CRISP) in Russell's viper. Our study identified the putative venom homologs from skin and the key residues that are changed across the members of Viperinae, Crotalinae and Elapidae that might have contributed towards the evolution of venom in vipers.

## MATERIALS AND METHODS

### Russell's viper shed skin and DNA isolation

Freshly shed skin of Russell's viper from Bangalore, India was a gift from Mr. Gerry Martin. The shed skin for the entire snake was obtained, cleaned thoroughly with 70% ethanol and with nuclease-free water three times each, dried thoroughly and frozen until the time of extraction of DNA. Genomic DNA was extracted following the protocol of *Fetzner Jr (1999)* with modifications.

### Sequencing, read processing and assembly

Illumina paired-end read libraries (100 base paired-end reads with insert size of 350 bases) were prepared following the manufacturer instructions using amplification free genomic DNA library preparation kit and sequenced using Illumina HiSeq2500 instrument. Archaeal, bacterial and human sequence contamination were removed from the Russell's viper sequence by DeConSeq (*Schmieder & Edwards, 2011*) using curated and representative genomes (https://www.ncbi.nlm.nih.gov/genome/browse/reference/).

Furthermore, the sequenced reads were post-processed to remove unpaired reads and quality analysis was performed using FastQC v0.1 (http://www.bioinformatics.babraham.ac.uk/projects/fastqc/). The rd_len_cutoff option was exercised during the read assembly step to trim off the low-quality bases, since the per-base quality was found to drop below 28 after the initial 50–70 bases of the read. The Russell's viper read libraries with $26\times$ coverage were assembled using SOAPdenovo2 (r240) (*Luo et al., 2012*).

### Identifying toxin gene homologs, coding regions, and predicted gene structures

The DNA sequences for 51 out of 54 venom-associated genes (*Fry, 2005*) from *Protobothrops mucrosquamatus* were downloaded (Table 1). These were used to fish genomic scaffolds bearing highly similar sequences in Russell's viper genome assembly, using BLAST with an $E$-value threshold of $10^{-3}$. The fished scaffolds were then anchored to the respective coding sequences from *Protobothrops mucrosquamatus*, using a discontiguous megaBlast, to determine the correct frame of translation and extract the complete amino acid coding sequence (CDS) corresponding to putative toxin homologs in Russell's viper. We obtained the exon-intron structures for all the putative toxin gene homologs in Russell's viper by aligning the CDS with gene sequences using discontiguous megaBlast and plotted using the tool GSDS2.0 (*Hu et al., 2015*). The sequences for the Russell's viper putative venom gene homologs were deposited in GenBank and their accession numbers are provided in Table S1.

**Table 1** **Genes and their representative families used in the current study.** The homolog with the highest identity was considered in cases with more than one homolog.

| Gene | Species with the available sequence information | Protein family |
|---|---|---|
| ACHE | Protobothrops mucrosquamatus, Ophiophagus hannah, Python bivittatus and Thamnophis sirtalis | Acetylcholinesterase |
| ADAM11 ADAM17 ADAM19 ADAM23 | Protobothrops mucrosquamatus, Ophiophagus hannah, Python bivittatus and Thamnophis sirtalis | ADAM (disintegrin/metalloprotease) |
| PROK1 | Protobothrops mucrosquamatus, Ophiophagus hannah and Python bivitattus | AVIT (prokinectin) |
| PROK2 | Protobothrops mucrosquamatus and Ophiophagus hannah, Python bivittatus and Thamnophis sirtalis | |
| CPAMD8 | Protobothrops mucrosquamatus, Python bivittatus and Thamnophis sirtalis | Complement C3 |
| crotasin | Protobothrops mucrosquamatus | Crotasin/beta defensin |
| CST1 | No sequence information is available in any of the four species | Cystatin |
| CST3 | Ophiophagus hannah | |
| CST4 | No sequence information is available in any of the four species | |
| CSTA | Protobothrops mucrosquamatus and Thamnophis sirtalis | |
| EDN1 | Protobothrops mucrosquamatus and Python bivittatus | Endothelin |
| EDN3 | Protobothrops mucrosquamatus, Ophiophagus hannah, Python bivittatus, Thamnophis sirtalis | |
| F5 | Protobothrops mucrosquamatus, Ophiophagus hannah, Python bivittatus, Thamnophis sirtalis and | Factor V |
| F10 | Ophiophagus hannah | Factor X |
| KLKB1 | Protobothrops mucrosquamatus, Ophiophagus hannah, Python bivittatus, Thamnophis sirtalis and | Kallikrein |
| KLK14 | Ophiophagus hannah | |
| kunitoxin | Protobothrops mucrosquamatus, Python bivittatus and Ophiophagus hannah | Kunitz-type protease inhibitor |
| LYNX1 | Ophiophagus hannah | LYNX/SLUR |
| CLEC3A | Protobothrops mucrosquamatus, Python bivittatus and Thamnophis sirtalis | Lectin |
| CLEC3B | Protobothrops mucrosquamatus, Ophiophagus hannah, Python bivittatus and Thamnophis sirtalis | |
| CLEC11A | Protobothrops mucrosquamatus, Python bivittatus and Thamnophis sirtalis | |
| CLEC16A | Protobothrops mucrosquamatus, Python bivittatus and Thamnophis sirtalis | |
| CLEC19A | Protobothrops mucrosquamatus and Python bivittatus | |
| NPR1 | Protobothrops mucrosquamatus, Python bivittatus and Thamnophis sirtalis | Natriuretic peptide |
| NPR2 NPR3 | Protobothrops mucrosquamatus, Ophiophagus hannah, Python bivittatus and Thamnophis sirtalis | |

| Gene | Species with the available sequence information | Protein family |
|------|--------------------------------------------------|----------------|
| NGF | *Protobothrops mucrosquamatus, Ophiophagus hannah, Python bivittatus, Thamnophis sirtalis, Protobothrops flavoviridis, Crotalus horridus, Sistrurus miliarius barbouri* and *Boa constrictor* | Beta-nerve growth factor |
| PLAA | *Protobothrops mucrosquamatus, Ophiophagus hannah, Python bivittatus* and *Thamnophis sirtalis* | |
| PLA2R1 | | |
| PLA2G1B | *Python bivittatus, Thamnophis sirtalis* and *Protobothrops mucrosquamatus* | |
| PLA2G10 | *Protobothrops mucrosquamatus, Protobothrops flavoviridis, Thamnophis sirtalis, Ophiophagus hanna* and *Python bivittatus* | |
| PLA2G12A | *Python bivittatus, Thamnophis sirtalis* and *Protobothrops mucrosquamatus* | |
| PLA2G12B | *Protobothrops mucrosquamatus, Ophiophagus hannah* and *Python bivittatus* | |
| PLA2G15 | *Protobothrops mucrosquamatus, Ophiophagus hannah, Python bivittatus* and *Thamnophis sirtalis* | Phospholipase A (2) |
| PLA2G3 | *Protobothrops mucrosquamatus, Python bivittatus* and *Ophiophagus hannah* | |
| PLA2G4A | *Protobothrops mucrosquamatus, Ophiophagus hannah, Python bivittatus* and *Thamnophis sirtalis* | |
| PLA2G4C | *Protobothrops mucrosquamatus, Python bivittatus* and *Thamnophis sirtalis* | |
| PLA2G6 | *Protobothrops mucrosquamatus, Ophiophagus hannah, Python bivittatus* and *Thamnophis sirtalis* | |
| PLA2G7 | | |
| SPSB4 | *Protobothrops mucrosquamatus* and *Thamnophis sirtalis* | |
| SPSB3 | *Protobothrops mucrosquamatus, Python bivittatus* and *Thamnophis sirtalis* | SPIa/Ryanodine |
| SPSB1 | *Protobothrops mucrosquamatus, Ophiophagus hannah, Python bivittatus* and *Thamnophis sirtalis* | |
| VEGFA1 | *Protobothrops mucrosquamatus, Ophiophagus hannah, Python bivittatus, Thamnophis sirtalis, Crotalus horridus* and *Protobothrops flavoviridis* | |
| VEGFA2 | | |
| VEGFA3 | | |
| VEGFB | *Protobothrops mucrosquamatus, Ophiophagus hannah, Python bivittatus, Thamnophis sirtalis, Crotalus horridus, Protobothrops flavoviridis* and *Sistrurus miliarius barbouri* | Vascular endothelial growth factor (VEGF) |
| VEGFC | *Protobothrops mucrosquamatus, Python bivittatus* and *Thamnophis sirtalis* | |
| VEGFF | *Protobothrops mucrosquamatus, Ophiophagus hannah, Python bivittatus, Thamnophis sirtalis* and *Protobothrops flavoviridis* | |
| WAP | *Protobothrops mucrosquamatus, Python bivittatus, Thamnophis sirtalis* and *Ophiophagus hannah* | Whey acidic protein/secretory leukoproteinase inhibitor |
| WFIKKN1 | | |
| WFIKKN2 | | |
| CRISP | *Protobothrops flavoviridis, Protobothrops mucrosquamatus, Ophiophagus hannah, Python bivittatus, Thamnophis sirtalis, Crotalus horridus, Calloselasma rhodostoma, Sistrurus miliarius barbouri* and *Deinagkistrodon acutus* | CRISP |

## Comparative analysis between venom proteins and their putative homologs

We obtained the accession IDs for the major toxin families from Russell's viper of the Indian sub-continent (Fig. S1 in *Sharma et al., 2015*). Their corresponding protein sequences were matched using blastp with the amino acid sequences from the putative skin homologs. For the genes covered under each family, a percent identity metric, indicative of the extent of sequence similarity between the venom proteins and their skin homologs, was estimated. Similar comparative analyses were performed for king cobra (*Ophiophagus hannah*) using accession IDs provided in Additional File 4 of *Tan et al. (2015)*, and the predicted toxin protein homologs from blood of king cobra (PRJNA201683 from *Vonk et al., 2013*). Comparative analyses were performed using blastp, with the venom protein sequence as the query, against PRJNA201683.

## Comparative analyses of putative venom protein homolog domains

The amino acid sequences of all the Russell's viper's putative toxin homologs were subjected to domain search using Pfam (*Finn et al., 2016*) (Table S2). All domain sequences were aligned using blastp to non-redundant protein sequences from 18 snake species (Table S3). We wanted to compare the gene structures between the venomous and the non-venomous animals, hence included sequence information from the members of the later group. Five domains (NGF, PDGF, Kunitz BPTI, CAP and CRISP) from four genes (NGF, VEGF, CRISP/Serotriflin, and Kunitoxin) with variability across different snake groups were used for expansive comparative analyses (Table S4). The sequences used were from viperids (taxid: 8689), elapids (taxid: 8602), colubrids (taxid: 8578), boids (taxid: 8572), acrochordids (taxid: 42164), pythonids (taxid: 34894), lizards (squamates (taxid: 8509) minus snakes (taxid: 8570), crocodiles (taxid: 51964) and testudines (taxid: 8459).

## 3D structure prediction of the chosen domains

Consensus sequences were determined from NGF, PDGF, Kunitz BPTI, CAP and CRISP domain alignments using Simple Consensus Maker (https://www.hiv.lanl.gov/content/sequence/CONSENSUS/SimpCon.html) for crotalines (CR), viperines (VP) and elapids. The consensus sequences were submitted to the protein fold recognition server (*Kelley et al., 2015*) using standard mode (http://www.sbg.bio.ic.ac.uk/phyre2/html/page.cgi?id=index). The best 3D model was further investigated by Phyre2 to analyze the structural model using various open source tools.

# RESULTS

## Shed skin yielded fairly good quality and high-molecular weight genomic DNA

Genomic DNA isolated from the shed skin of Russell's viper was fairly intact with most of the DNA in the size range of more than 5 kbp (Fig. S1). Sequenced short reads were assembled and then used to fish the sequences for the 51 putative toxin genes in Russell's viper (see 'Materials and Methods'). Next, we obtained the exon-intron structures for all putative homologs in Russell's viper by aligning the CDS with gene sequences (Fig. S2).

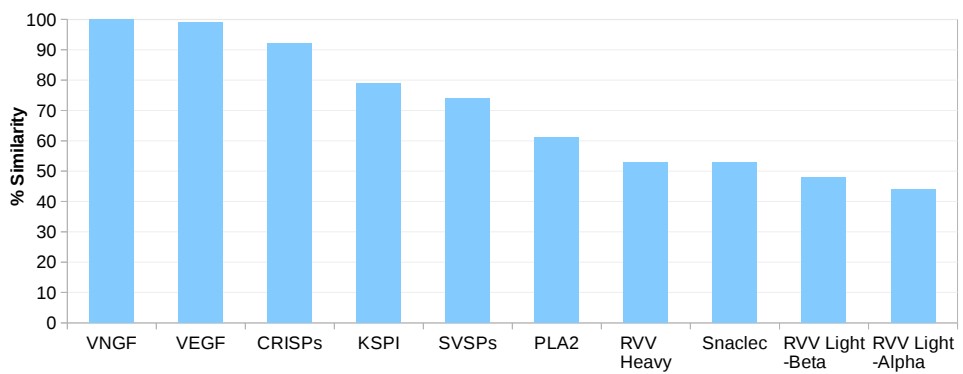

**Figure 1** Sequence identity (%) between the proteins from ten major venom families and their putative skin homologs in Russell's viper. The homolog with the highest identity was considered where more than one homolog was present.

We found the average length of the exons in those sequences to be around 190 nucleotides (nt), matching well with the lengths of other vertebrate exons (*Gelfman et al., 2012*).

## Similarity between venom proteins and their putative skin homologs

For the Russell's viper, we found 45–100% sequence similarity between the major venom proteins and their predicted putative skin homologs (Fig. 1). The sequences for venom nerve growth factor (VNGF) and its putative skin homolog were identical. Similarly, VEGF and CRISPs from venom gland were highly similar to their putative skin homologs (99% and 92% sequence similarity, respectively). Other proteins like, KSPI, SVSPs and PLA2 showed 79%, 74% and 61% sequence identity, respectively (Fig. 1 and Fig. S3). In order to find out whether the sequence divergence between some of the venom gland proteins and their predicted putative skin homologs was specific to Russell's viper, we performed similar analysis using venom proteins and their blood homologs from king cobra, *Ophiophagus hannah* (*Vonk et al., 2013*). In the case of *Ophiophagus hannah*, the differences between toxin proteins and their blood homologs were minor for most families studied (similarity ≥75%), except for PLA2, which had a low similarity of 23% (Figs. S4 and S5).

## Comparative domain analyses

Among the genes, a larger pool of sequences were available only for NGF, PDGF domain of VEGF, Kunitz_BPTI domain of Kunitoxin, CRISP and CAP domains in CRISP and Serotriflin proteins, from various snake groups (Colubridae, Boidae, Pythonidae and Acrochordidae), non-snake reptilian groups (lizards, crocodiles and Testudines), venomous invertebrates (wasps, spiders and scorpions) and venomous vertebrates (fishes and mammals). Therefore, these domains were compared with their putative homologs from Russell's viper. Comparative domain analysis was performed for all putative toxin gene homologs (Fig. S6) across 18 snake species for those where sequence information was available (Table S3). In the case of five domains: CAP and CRISP domains of CRISP and serotriflin genes (L and AL), Kunitz BPTI of kunitoxin (S), NGF (T) and PDGF of VEGFA (AP-AR) and VEGFF (AU), we found that the maximum number of species aligned to

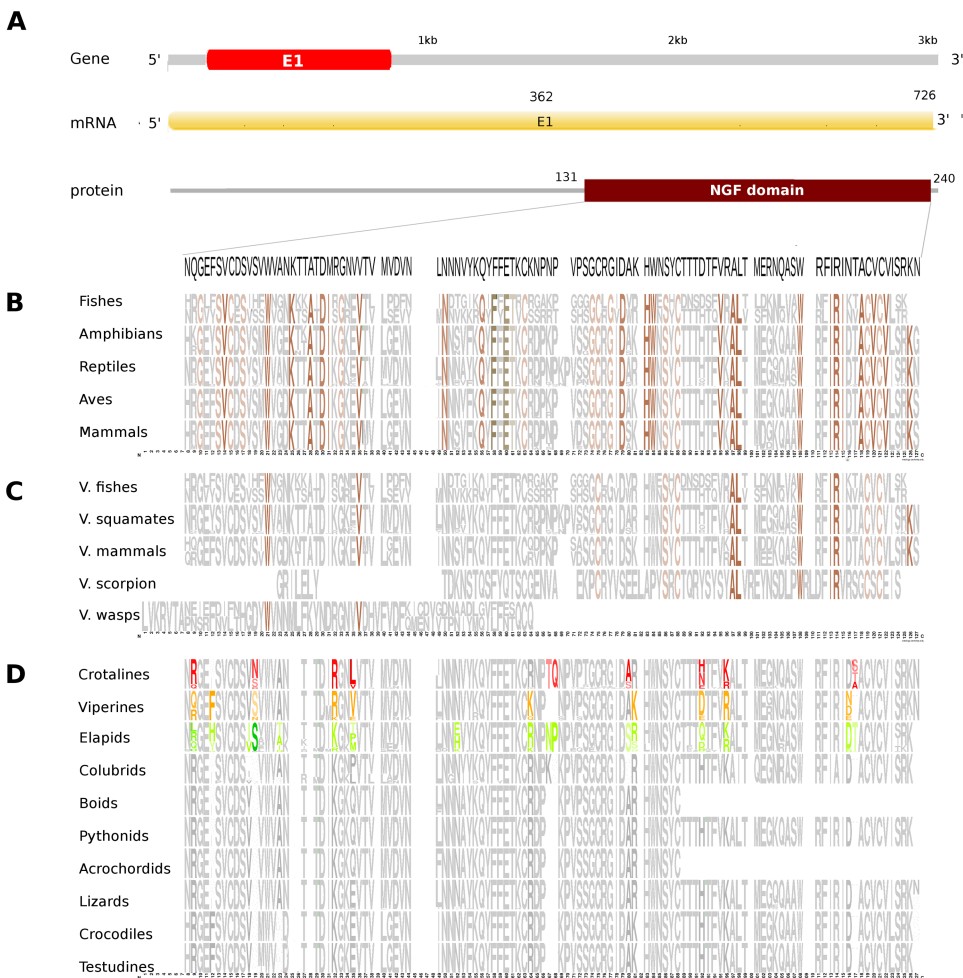

**Figure 2** **Comparative analyses of nerve growth factor (NGF).** Putative NGF gene homolog, its mRNA, and protein domains in Russell's viper (A) and its comparison with the consensus NGF sequences from all five vertebrate phyla (fishes, amphibians, reptiles, birds and mammals) (B), with venomous (V) vertebrates from multiple phyla of vertebrates and invertebrates (C), and from various reptilian subgroups (D) are shown. The shades of brown and grey in (B) and (C) represent conservation to various degrees and variability, respectively. Grey in (D) represents conserved residues, red represents variable residues in the crotalines (CR), yellow and green represent variable residues in the viperines (VP), and elapids respectively.

their domain sequences. Some protein domains, the CRISP, Kunitz BPTI, guanylate CYC, PDGF of VEGFF and WAP, showed long stretches of mismatches (Fig. S6) compared with Russell's viper sequence. Out of these, only NGF and PDGF domains of VEGF had amino acid changes specific to the members of the group Crotalinae, that were completely absent in any other group used for comparison, including in lizards, crocodiles, and turtles (Fig. S7). Specific changes in these proteins and their implications are discussed below.

The putative skin-derived NGF gene homolog in Russell's viper is a single exon gene with a 745nt transcript coding for a 244 amino acid protein consisting of a single NGF domain (Fig. 2A). The NGF domain bears 28% sequence conservation across all the five vertebrate

phyla, namely, fishes, amphibians, reptiles, aves and mammals distributed along the length of the domain (Fig. 2B). Thirty-six percent out of these residues are conserved across other venomous vertebrates (fishes, squamates and mammals) and venomous invertebrates (scorpions and wasps) (Fig. 2C). Thirteen percent of the skin-derived putative NGF domain residues are variable with respect to the domain sequence in at least one among the NGF sequences in the groups of viperids and elapids (Fig. 2D). Although several amino acids in the NGF domain in crotalines seem to have changed from the putative domain in Russell's viper and other vipers of the group Viperinae, their function probably remains unchanged. For example, phenylalanine (F) to isoleucine (I) at position 12 and serine (S) to asparagine (N) at position 19 between the crotalines and viperines does not change the function of the amino acids (from one hydrophobic amino acid to another and from one polar amino acid to another). However, it is also true that F changing to I removes the bulky aromatic ring, whereas S could be a phosphorylated site as opposite to the N in the same position. There are others, for example, threonine (T) and glutamine (Q), at positions 67 and 68, respectively, in the NGF domain of the crotalines, which were only there in that specific group. One of those, a polar amino acid glutamine at position 68, is a very important residue as its corresponding amino acid in any of the other snakes, except in colubrids, is a hydrophobic proline.

In Russell's viper, the putative skin-derived VEGFA gene homolog comprises five exons coding for a 652 nt long transcript and a translated protein with two domains: PDGF and VEGF-C (Fig. 3A). The PDGF domain sequence exhibits conservation in 65% of its residues across the three vertebrate phyla (reptiles, birds and mammals) (Fig. 3B). Since sequence information from fishes and amphibians were not available, we could not include those in the comparison study. Out of the conserved residues in the above said domain, 21% were also conserved in venomous vertebrates (squamates and mammals) and venomous invertebrates (wasps). Fifteen percent of the PDGF domain residues were variable in at least one of the two snake groups: viperids and elapids (Figs. 3C and 3D). Like the NGF domain, the evolution of the putative skin-derived PDGF domain in crotalines at certain amino acids is striking. For example, in the crotalines, the position 67 is a polar amino acid tyrosine (Y) while in all other reptiles, venomous invertebrates and mammals; this is primarily a hydrophobic amino acid phenylalanine (F).

The putative skin-derived Kunitoxin gene homolog in Russell's viper is a 3.1 kb gene comprising two exons, with a transcript length of 270 nt that codes for a 44 amino acids long single Kunitz BPTI domain (Fig. 4A). About 29% of the protein domain residues are conserved across the four vertebrate phyla (amphibians, reptiles, aves and mammals) (Fig. 4B). Since sequence information from the Kunitz BPTI for fishes was not available, we could not include those in the comparison. Out of these conserved residues, 76% are conserved in venomous vertebrates (squamates and mammals) and venomous invertebrates (scorpions and wasps) (Fig. 4C) and 56% of the domain residues are variable in at least one of two snake groups (viperids and elapids) (Fig. 4D). Of the residues that are evolved in the members of Crotalinae, the second residue, a positively charged one, arginine (R) is present only in the members of Viperinae, which is replaced by a hydrophobic residue,

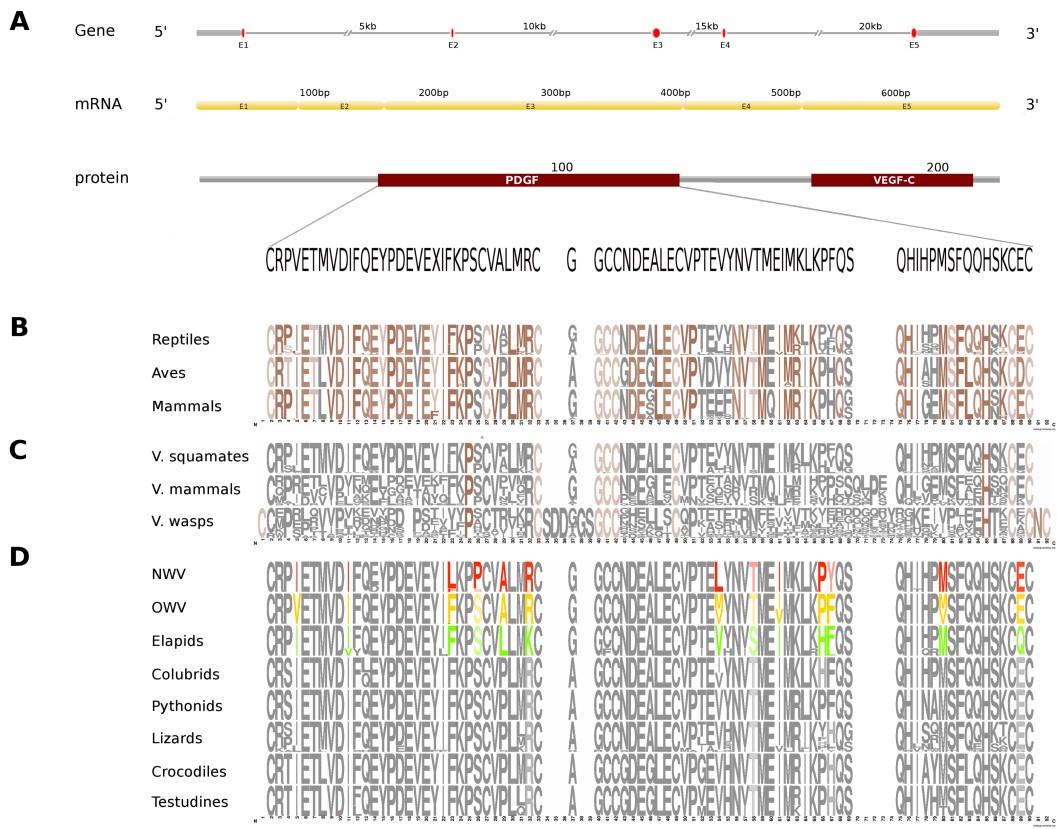

**Figure 3** **Comparative analyses of vascular endothelial growth factor—A (VEGF-A).** Organization of the putative gene homolog, its mRNA, and protein domains of Russell's viper PDGF domain (A) and its comparison with the consensus sequences from all five vertebrate phyla (fishes, amphibians, reptiles, birds and mammals) (B), from the venomous (V) vertebrates and invertebrates (C), and from various reptilian subgroups (D) are shown. The shades of brown and grey in (B) and (C) represent conserved and varying residues, respectively. Grey in (D) represents conserved residues, red represents variable residues in the crotalines (CR), yellow and green represent variable residues in viperines (VP), and elapids respectively.

proline (P), in the crotalines and elapids. Residues 14–18 are very polymorphic in the crotalines and elapids, but not so in the viperines.

The skin-derived putative CRISP gene homolog in Russell's viper is a 25 kb long gene, comprises of eight exons coding for a 787 nt transcript and two translated protein domains, CAP and CRISP (Fig. 5A). The CAP domain exhibits conservation in 7% of its residues across all the five vertebrate phyla (Fig. 5B). Forty-two percent of those residues are conserved across venomous vertebrates (amphibians, squamates and mammals) and venomous invertebrates (scorpions and wasps) (Fig. 5C). In addition, there are five residues conserved across all the venomous animals (Fig. 5C). Twenty-seven percent of the CAP domain residues are variable in at least one of the three snake groups (Fig. 5D). There are several extra residues for the CAP domain in the crotalines and elapids, but not in the viperines. The conserved residues comprised mostly of Cystines and to a lesser extent Asparagines (Fig. 5E) across venomous vertebrates (squamates and mammals) (Fig. 5F).

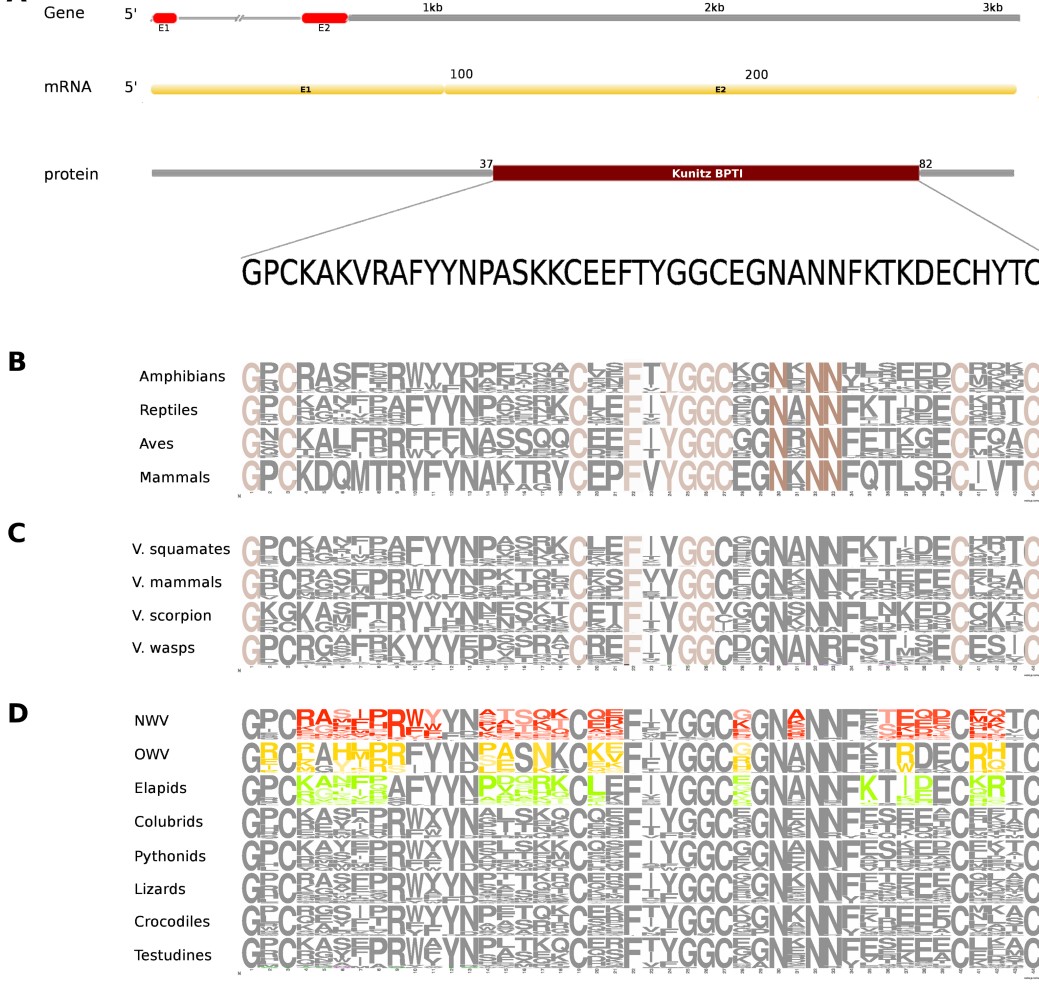

**Figure 4 Comparative analyses of kunitoxin.** Organization of the putative gene homolog, its mRNA, and protein domains of Russell's viper (A) and its comparison with the consensus BPTI domain sequences from all five vertebrate phyla (fishes, amphibians, reptiles, birds and mammals) (B), from venomous (V) vertebrates and invertebrates (C), from various reptilian subgroups (D) are shown. The shades of brown and grey in B and C represent conserved and varying residues, respectively. Grey in D represents conserved residues, red represents variable residues in the crotalines (CR), yellow and green represent variable residues in viperines (VP), and elapids respectively.

Sixty percent of the residues in the putative homolog of the CRISP domain are variable in at least one viperine or elapid member with respect to the domain sequence of Russell's viper (Fig. 5G).

Next, we explored the role of consensus domain sequences in the putative protein homologs and the possible role of conserved amino acids in those domains across viperids and elapids. We constructed the 3D structure models using Phyre2, followed by Phyre2 investigation, for further analyses on the structural model. As evident from the analyses, amino acid residues 18–19 and 117 of the NGF domain reflected a difference in mutation sensitivity as detected by SusPect algorithm (*Yates et al., 2014*), especially in the elapids

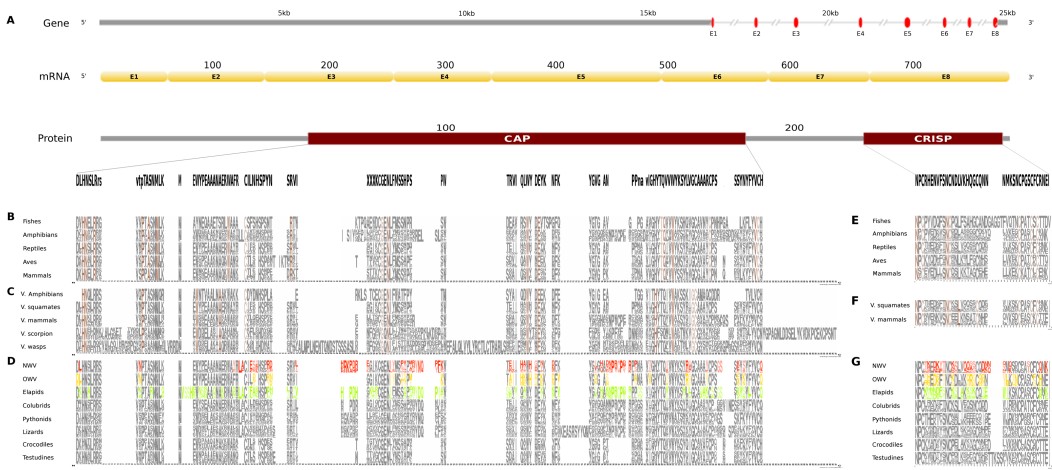

**Figure 5** **Comparative analyses of CRISP.** Organization of putative CRISP gene homolog, its mRNA, and protein domains of Russell's viper (A) and its comparison with the consensus CRISP sequences from all five vertebrate phyla (fishes, amphibians, reptiles, birds and mammals, B and E); from venomous animals (V) vertebrates (fishes, squamates and mammals) and invertebrates (scorpions and wasps, C and F); and from various reptilian subgroups (D and G) are shown. The shades of brown and grey in B, C, E and F represent conserved and varying residues, respectively. Grey in D and G represents conserved residues, red represents variable residues in the crotalines (CR), yellow and green represent variable residues in viperines (VP), and elapids respectively.

compared to the viperids (Fig. 6). There were differences in certain residues across these two groups. Residue 18 is Valine in the viperines and Isoleucine in the elapids; residue 19 is Serine in the viperines and Asparagine in the crotalines; and residue 117 is Threonine in the elapids and Serine in the crotalines (Fig. 6A). This might have implications in the structure of the protein as the largest pockets detected by fpocket algorithm appear to be vastly different among the crotalines, viperines and elapids for the NGF, PDGF, CAP and CRISP domains (Fig. 6). The pockets appeared smallest in all cases for the elapids (quantitatively substantial for PDGF, CAP and CRISP: one-fourth that of viperines for PDGF, and two-thirds that of viperines for CAP and CRISP), and largest in the case of viperines (Fig. 6). Minor differences in clashes were observed at residues 10, 11 and 20 of the Kunitz domain and residue 38 of this domain showed a rotamer conflict in the case of the crotalines (Fig. 6C). Similarly, residue 46 of the CAP domain and residues 4 and 31 of the CRISP domain showed rotamer conflict for the viperines (Figs. 6D and 6E). The other protein quality and functional parameters were not affected across the 3D structure models for the three snake groups (Fig. S8).

## DISCUSSION

Accessibility and affordability of high-throughput sequencing technologies along with the availability of sophisticated computational tools to assemble, annotate and interpret genomes is playing a powerful role in deciphering gene functions and their role in evolution. Snake toxin genes are coded by gene families and produce gene isoforms through the process of duplications (*Casewell et al., 2013*; *Fry, 2005*). Several studies on

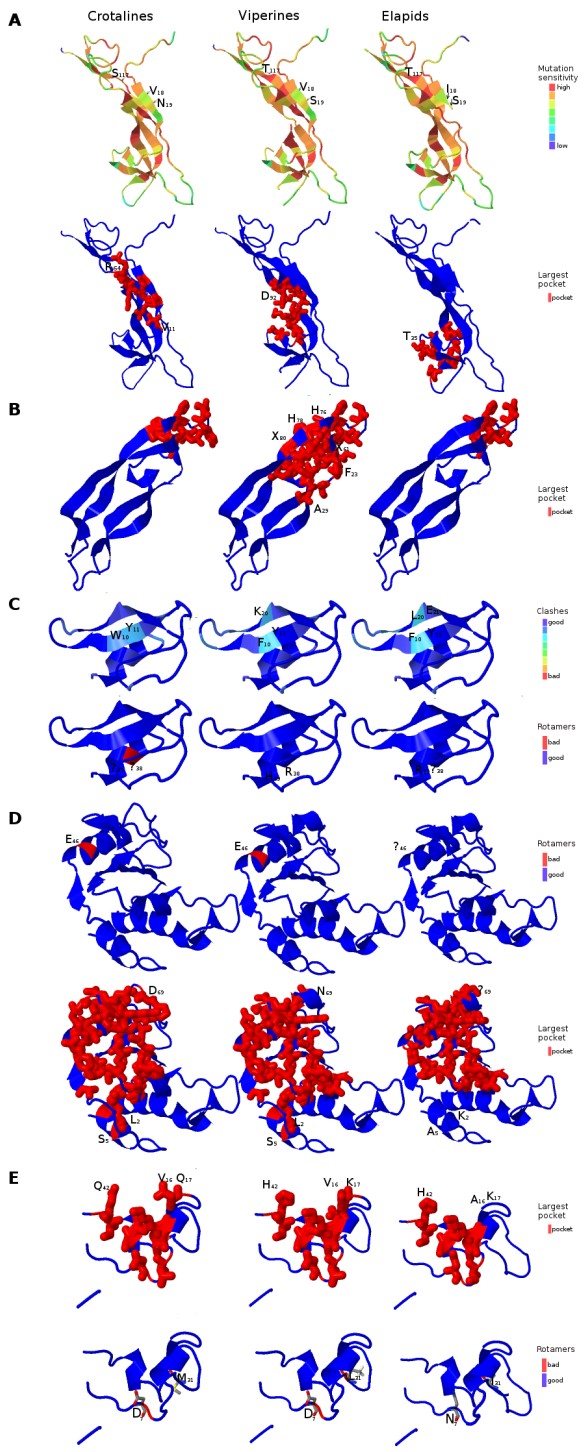

**Figure 6 Three dimensional protein structural models in NGF (A); PDGF (B); Kunitz BPTI (C); CAP (D); and CRISP (E) across crotalines (CR), viperines (VP) and elapids.** The status of the parameters being investigated using Phyre2 are indicated in the color legends on the side.

the venom-associated proteins from New World vipers have classified the venoms into four groups (type I–IV), based on the relative abundance of toxin families (*Calvete, 2013*; *Gibbs et al., 2013*; *Goncalves-Machado et al., 2016*; *Jimenez-Charris et al., 2015*; *Lomonte et al., 2014*; *Mora-Obando et al., 2014*; *Pla et al., 2017*; *Salazar-Valenzuela et al., 2014*). The different groups are: snake venom metalloproteinase-predominant (type I), heterodimeric β -neurotoxic PLA2 –rich (type II), serine proteinases and PLA2 (type III) and type IV, which is similar to type III but with significant higher concentration of snake venom metalloproteinases (*Calvete, 2017*). Russell's viper (*Daboia russelii*) is a Old World pitless viper, characterized by the lack of heat sensing pit organs (*Mallow, Ludwig & Nilson, 2003*).

There is significant variation in the venom composition of Russell's viper in India (*Jayanthi & Gowda, 1988*; *Sharma et al., 2015*), making the universal anti-venom less effective against all Russell's viper bites across the country. The variation in the venom composition within the same species is thought to be a result of adaptation in response to the difference in diets (*Barlow et al., 2009*; *Casewell et al., 2013*; *Daltry, Wuster & Thorpe, 1996*). A comparison across four published studies (*Kalita, Patra & Mukherjee, 2017*; *Mukherjee, Kalita & Mackessy, 2016*; *Sharma et al., 2015*; *Tan et al., 2015*) on Russell's viper venom proteins revealed that the composition of some of the major venom proteins varied significantly (Fig. S9). For example, in one study (*Mukherjee, Kalita & Mackessy, 2016*), VNGF constituted only 0.4% of the venom while in another (*Kalita, Patra & Mukherjee, 2017*), the same protein constituted 4.8% of the venom. As both studies came from the same lab, there is little chance for any technical or assay-related variability. In the first study, the venom was used from the captive species in a zoo in the USA where the snake was from a Pakistani origin (*Mukherjee, Kalita & Mackessy, 2016*) while the other study used venom from a commercial source in India (*Kalita, Patra & Mukherjee, 2017*). This suggests that there is a great deal of variation in the composition of Russell's viper venom collected from different locations, corroborating the earlier results (*Jayanthi & Gowda, 1988*; *Sharma et al., 2015*). Currently, efforts are underway to collect venoms of Russell's viper from different regions of India in order to understand their composition (Rom Whitaker & Gerry Martin, pers. comm. with Binay Panda, 2017).

Studies on venom-associated genes using whole-genome sequencing data in Russell's viper are scarce. One of the reasons is the relative difficulty in accessing venom glands from snakes. This can be partially addressed by studying their putative homologs from shed skin, which is relatively easy to access. Past studies on the members of Viperidae focused on proteins and used proteomics-based analyses (*Gao et al., 2013*; *Gao et al., 2014*; *Kalita, Patra & Mukherjee, 2017*; *Li et al., 2004*; *Liu et al., 2011*; *Mukherjee, Kalita & Mackessy, 2016*; *Sharma et al., 2015*; *Tan et al., 2017*; *Tan et al., 2015*; *Villalta et al., 2012*). The only viperine where complete genome sequence information is available is a European adder, *Vipera berus berus* (https://www.ncbi.nlm.nih.gov/bioproject/170536). Although sequence information is available for this species, the annotation is not available and therefore could not be used in our study.

The aim of the current study was two fold. First, as handling and getting biological material from snakes requires specific expertise, we wanted to test whether one can obtain high-molecular weight DNA from shed skin as a source of analyte for genome sequencing.

Second, we wanted to study the potential of skin-derived putative toxin gene homologs, as surrogates of their venom gland counterparts through comparative analyses. On the first account, we found the results to be satisfactory. Although shed skin is often contaminated with bacteria and other microorganisms, and the DNA obtained from the shed skin is sheared, we show that one can successfully isolate high-molecular weight genomic DNA from shed skin (Fig. S1). Therefore, shed skin may be an attractive option in the future for generating snake genome data. We showed that the comparisons of amino acid sequence and three dimensional structures of five toxin domains with their putative skin homologs across the major kingdoms of life can generate important information towards understanding the macro- and micro-evolution of these genes. Results from our comparative analyses showed that some of the venom gland proteins are identical or near identical to their putative skin-derived homologs (VNGF, 3FTX and LAAO) but others had low overall similarity (Snaclec and RVV). We were curious to find out whether the low sequence similarity for some venom proteins with their putative homologs was specific to Russell's viper and how much of the low overall similarity in those proteins was due to the heterogeneity, if any, found among snakes of the same species. Comparative analysis between the toxins and their blood homologs in king cobra provided us with an answer for the first question where 7 out of 8 venom proteins studied (except for PLA2) were very similar to their blood homologs (Figs. S4 and S5). This suggests that some venom proteins may not be that different from their homologs in other organs. A recent study in python, where the authors argue that the functional evidence of toxic effects on prey and not their expression is the correct criterion to classify proteins as venom toxins (*Reyes-Velasco et al., 2015*), strengthens this hypothesis further. However, we are aware that this may vary from species to species and in some species the venom proteins may be very different from their homologs in other organs.

In our study, we compared venom proteins described previously (*Sharma et al., 2015*) using animals captured near Chennai, Tamil Nadu, India with their skin-derived putative homologs from a completely different animal (shed skin was collected in Bangalore, Karnataka, India). The distance between these two places is roughly 350–400 km. Therefore; it is possible that in our study, the low similarity in some of the venom proteins with their putative skin homologs could have been due to the variation in the venom composition of the animals in these two locations. Despite this, 50% of the venom proteins studied had >75% and 3 had near perfect sequence similarity with their putative skin homologs. A clear picture will emerge from a direct comparison between the venom proteins and their skin counterparts from the same animal.

From the sequence data, we succeeded in assembling near complete CDS for 20 gene families representing 51 gene homologs (*Fry, 2005*). This highlights the utility of genome sequencing data in inferring putative toxin gene homologs. As the lengths of the putative toxin gene homologs in Russell's viper were much longer than the CDS, the intronic sequences were assembled with gaps. This was primarily due to the low coverage sequencing data used for assembly and the lack of long-insert mate pair sequencing data in our repertoire. The aim of the current study was not to assemble a perfect genome for Russell's viper but to use the low coverage data to fish out putative toxin gene homologs
from skin. The mean length of exons for the putative toxin gene homologs in Russell's viper was 190 bases, much smaller compared to the average intron length. In our study, we could assemble exons accurately using short-read sequence data. Interestingly, we found that the AT to GC ratio in the CDS regions (cumulatively for all the 51 toxin gene homologs) was 1:1 whereas it was skewed (the ratio is 1.5:1) for the full gene sequences.

Our study demonstrates the feasibility of *de novo* sequencing and analyses of gene families without prior sequence information and annotation, and without going through the process of designing individual primers for Sanger sequencing. However, there are certain limitations to our study. First, it focuses on the putative toxin gene homologs from skin and not the toxin genes from venom gland. There is a possibility that the toxin genes from the same animal in the venom gland are different from their homologs in the skin, and therefore can only be described as putative. Hence, we can neither be sure of the presence nor the activity of the homologs in the skin. Although the skin-derived transcriptome data will add value to the study, it will still be inadequate. Future transcriptome and proteomics analysis from both the skin and the venom gland, preferably from the same animal, along with their functional studies will only be definitive. Second, like any other annotation-based study, we relied on the quality of existing/prior annotation of toxin-related genes. It is possible that due to sequencing artefacts, there are errors in the assembled genomic sequence, and therefore in the translated protein sequences inferred in our study. Future studies using high-coverage sequencing data along with data on RNA and protein to derive better gene annotations along with the functional studies on their spatial and temporal expression will point to the true functional significance of skin-derived toxin gene homologs.

## ACKNOWLEDGEMENTS

We thank Kunal Dhas and Arun Hariharan for their help in laboratory experiments.

### Funding

This work was supported by Department of Electronics and Information Technology, Government of India [18(4)/2010-E-Infra., 31-03-2010]; and Department of IT, BT and ST, Government of Karnataka, India [3451-00-090-2-22]. The funders had no role in study design, data collection and analysis, decision to publish, or preparation of the manuscript.

### Grant Disclosures

The following grant information was disclosed by the authors:
Department of Electronics and Information Technology, Government of India: [18(4)/2010-E-Infra, 31-03-2010].
Department of IT, BT and ST, Government of Karnataka, India: [3451-00-090-2-22].

### Competing Interests

The authors declare there are no competing interests.

## Author Contributions

- Neeraja M. Krishnan and Binay Panda conceived and designed the experiments, performed the experiments, analyzed the data, contributed reagents/materials/analysis tools, wrote the paper, prepared figures and/or tables, reviewed drafts of the paper.

## DNA Deposition

The following information was supplied regarding the deposition of DNA sequences:

Russell's viper sequence data is deposited in the NCBI SRA database under the accession number SRR5506741 and the individual gene sequences are deposited in the GenBank (accession numbers in Table S1).

## Data Availability

GeneBank SRA, accession number SRR5506741.

## Supplemental Information

Supplemental information for this article can be found online at http://dx.doi.org/10.7717/peerj.4104#supplemental-information.

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
