# Peer review of "Comparative analyses of putative toxin gene homologs from an Old World viper, Daboia russelii"

_PeerJ, doi:10.7717/peerj.4104_

## Round 0.1 · original submission · Major Revisions

Please pay close attention to the comments of both reviewers. I hope that you will be able to address all critical points.

Reviewer 1 ·

Basic reporting

Although the English has been improved from the previous version of the same paper, which I also reviewed, there is still some repetition between sections. Zoological details are also frequently inaccurately given: eg Protobothrops mucrosquamatus is not a New World pitviper; on line 101, the word “crotalin” should be “crotaline” while on line 230, the word “crotalinae” should be capitalised. Line 331: the correct spelling of the Latin name of Russell’s viper is Daboia russelii. Line 345: correct spelling is Viperidae. There are too many figures of dubious value (even if the analysis was considered sound, for which see following sections).

Experimental design

In a previous review of this study for another journal, I pointed out that while a major comparison made in the paper was between "Old World vipers" and "New World vipers", crotalines are present in the Old World too, and it was unclear whether they were being included in the OWV category, or whether the comparison was in fact between the Viperinae and the Crotalinae subfamilies (since all the NWV are crotalines). The authors have amended the analysis to apparently compare crotalines and viperines instead, but they now describe Protobothrops mucrosquamatus as a New World pitviper, which it is not (note also that I also doubt that any re-analysis has actually be done since the results reported for the comparative analysis are identical (word for word) to the previous version.

Validity of the findings

Although the units being compared have apparently been improved, compared to the previous version, and the wording slightly changed so that they no longer claim the “fished” sequences to be “venom-associated” as they did in the former version, there are still some unjustified assumptions being made which invalidate the analysis. Foremost among these is that the genes fished out of genomic DNA extracted from the skin have anything to do with expression in the skin…they do not. Without transcriptomic studies, we have no idea where they are expressed. They may be expressed in the venom gland and they may be expressed in the skin, but could also be unexpressed or expressed all over the body. This invalidates all the conclusions being drawn and I therefore will not comment on them further.

Additional comments

Although the authors have deleted a comment suggesting that conventional sampling can adversely affect the snakes from this version as requested by me in my previous review, I have recently become aware that according to Indian Wildlife Protection Act, 1972, and its subsequent amendments, even the shed skin (not exuviate, which is an incorrect term as I have already pointed out) is the property of the Government. Hence, it does not really circumvent the need to obtain permits. Nevertheless the ability to recover high quality DNA from the shed skin is certainly worthy of a note in a journal focusing on methodological studies. However, additional details will be needed such quality of the assemblies (N50, % coverage etc.).

As I have already suggested in my previous review, it would perhaps be of interest to do a phylogenetic analysis to see if those encoding proteins that have been found in the venom of Russell's viper or related spp in the past (ie from EST or transcriptomic studies) for which the tag "venom- associated" can be justified, are more closely related to each other than to non-toxic housekeeping genes (e.g. Jacobo Reyes-Velasco, Daren C. Card, Audra L. Andrew, Kyle J. Shaney, Richard H. Adams, Drew R. Schield, Nicholas R. Casewell, Stephen P. Mackessy, Todd A. Castoe; Expression of Venom Gene Homologs in Diverse Python Tissues Suggests a New Model for the Evolution of Snake Venom. Mol Biol Evol 2015; 32 (1): 173-183. doi: 10.1093/molbev/msu294; Casewell NR, Huttley GA, Wuster W. Dynamic evolution of venom proteins in squamate reptiles, Nat Commun. 2012, vol. 3 pg. 1066). However, you would need to bear in mind that you just have coding sequence data, which will be confounded by strong selection pressure (see Malhotra A, Creer S, Harris JB, Thorpe RS. 2015. Toxicon 107(Pt B):344-58). The noteworthy finding in this work is that DNA suitable for NGS can be obtained from shed skin and the report ought to focus on this aspect, but probably additional analyses would be needed to compare the quality with other more standard sources.

Reviewer 2 ·

Basic reporting

no comment

Experimental design

no comment

Validity of the findings

no comment

Additional comments

When the residue identity was compared and discussed, the authors seemed to use hydrophobic vs polar as the only criteria, which is useful but not complete. For instance, in figure 2 and page 16, F changing to I removes the bulky aromatic ring, whereas S could be a phosphorylated site as opposite to the N in the same position. This could bring bigger difference in the protein structure and function than just changing the polarity. The authors need to address this to improve the argument.

In figure 2, residues 67 and 68 don’t align with the original sequence (above figure 2B). It is hard to figure out the corresponding counterpart residue. The author can either change the font of the sequence or point out simply by adding arrow. Same issue in figure 6A, it is not clear to pinpoint residue 117, 18 and 19. Adding an arrow to point to the secondary structure would be helpful.

In figure 4 and page 18, ’ Of the residues that are evolved in the members of crotalinae, the second residue, a positively charged one, alanine (A) is present only in the members of viperinae, which is replaced by a hydrophobic residue, proline (P), in the crotalines and elapids’, which is the positively charged residue? Ala is not charged. Is this supposed to be Arg (R)?

In figure 6 (B and D) and page 19 ’The pockets appeared small in all cases for the elapids, and largest in the case of viperines’. It would be useful and straightforward to quantitatively describe the size of pocket.

In figure 6 E and page 19, ’ Similarly, residue 46 of the CAP domain and residues 4 and 31 of the CRISP domain showed rotamer conflict for the viperines’. Residue 4 is in the loop region, which could experience highly dynamics motions and might be different rotamer configuration. How significant of this? Does different rotamer configurations imply anything in the protein function or it is purely due to the dynamics in the loop. It would be useful to show the side chain in the figure to clearly display the rotamer conflict between different speicies.

Several references are missing in the reference section. For instance, in page 23, ‘..When we compared the 4 published studies (Kalita et al. 2017; Mukherjee et al, 2016; Sharma et al. 2015; Tan et al. 2015)…’ Kalita et al. 2017; Mukherjee et al, 2016 and Tan et al. 2015 are not listed in the literature cited. This happened in other places too and the literature cited is very short considering the reference talked in the maintext. The authors need to read full text carefully and add the corresponding references accordingly in the literature cited section.

---

## Round 0.2 · accepted · Accept

Special thanks for the detailed response to the reviewers' comments for careful revision of the manuscript.